# Ubiquitin-Activating Enzyme E1 (UBA1) as a Prognostic Biomarker and Therapeutic Target in Breast Cancer: Insights into Immune Infiltration and Functional Implications

**DOI:** 10.3390/ijms252312696

**Published:** 2024-11-26

**Authors:** Mingtao Feng, Huanhuan Cui, Sen Li, Liangdong Li, Changshuai Zhou, Lei Chen, Yiqun Cao, Yang Gao, Deheng Li

**Affiliations:** 1Department of Neurosurgery, Fudan University Shanghai Cancer Center, Shanghai 200032, China; 13601728046@163.com (M.F.); cuihuanhuan@21cn.com (H.C.); joshuasen666@163.com (S.L.); lild0123@163.com (L.L.); shiwa1990@163.com (C.Z.); 861025yueyang@163.com (L.C.); yiqun_fduscc@163.com (Y.C.); 2Department of Oncology, Shanghai Medical College, Fudan University, Shanghai 200032, China

**Keywords:** UBA1, breast cancer, prognostic biomarker, immune infiltrates

## Abstract

Ubiquitin-Activating Enzyme E1 (UBA1), an E1 enzyme involved in the activation of ubiquitin enzymes, has been involved in the onset and progression of different cancers in humans. Nevertheless, the precise contribution of *UBA1* in breast cancer (BC) is still poorly characterized. In this study, a thorough investigation was carried out to elucidate the significance of UBA1 and validate its functionality in BC. Through the analysis of mRNA sequencing data of BC patients, the mRNA expression of *UBA1* was observed to be notably enhanced in cancer tissues relative to controls, and high *UBA1* expression was linked to worse overall survival (OS), disease-specific survival (DSS), and progress-free survival (PFS). Moreover, *UBA1* exhibited potential as an independent prognostic and diagnostic biomarker for individuals with BC. Additionally, functional enrichment analysis revealed the involvement of *UBA1* in inflammation-linked pathways, like the TNF-α signaling pathway, the IL-6 signaling pathway, and various immune-related biological processes. Notably, single-sample gene set enrichment analysis (ssGSEA) aided in the identification of a negative link between *UBA1* expression and the levels of infiltrating mast cells, Th1 cells, iDC cells, B cells, DC cells, Tem cells, Cytotoxic cells, T cells, CD8T cells, and pDC cells. Finally, this study demonstrated that silencing UBA1 significantly impeded the growth and development of BC cell lines. These findings highlight UBA1 as a potential prognostic biomarker linked to immune infiltration in BC, thereby depicting its potential as a new therapeutic target for individuals with BC.

## 1. Introduction

Breast cancer (BC) is the most prevalent malignancy among females, representing a leading cause of death due to cancer. In recent years, there has been a global increase in the incidence of BC, thereby posing a significant threat to women’s health and well-being and placing a substantial burden on healthcare systems [1]. As of 2022, around 4.1 million females in the United States had a history of BC, with 4% of these patients experiencing cancer metastasis [2]. BC is a heterogeneous disease, and its occurrence, progression, and prognosis are influenced by various factors, including genetics and the environment. However, the core mechanisms driving BC pathogenesis still need to be comprehended.

Recent advancements in diagnostic and treatment technologies, such as *BRCA1/BRCA2* mutation analysis and targeted therapies against HER2, PD-1, and PD-L1, have significantly improved the prognosis of BC patients [1,3,4]. Nonetheless, accurately identifying patients at risk of treatment failure and understanding the specific resistance mechanisms involved continue to be challenging due to the pronounced heterogeneity of BC. Additionally, predicting the most effective treatment modality for individual patients remains a formidable task. Thus, further investigations into the molecular mechanisms underlying BC progression and the exploration of new biomarkers and therapeutic targets are warranted.

Ubiquitination, an important post-translational modification of proteins, significantly contributes to various physiological processes, such as apoptosis, cell proliferation, metabolism, immune response, and differentiation [5]. Ubiquitination also regulates the pathways of tumor occurrence and inhibition. Mutated proteins in tumors can be degraded through ubiquitination, and dysregulated ubiquitination mechanisms can alter the activity of tumor suppressors [6]. Numerous studies have shown the involvement of ubiquitination in BC progression [7]. For instance, Cho Y et al. demonstrated that the ubiquitination of stem factor *OCT4* in BC stem cells can inhibit the development of BC [8]. Zhu J et al. identified the atypical E3 ubiquitin ligase RNF31 as a potential therapeutic target for BC patients, as it is highly expressed in BC and stabilizes estrogen receptor α (ER α) while participating in p53 degradation [9,10]. Moreover, targeting the ubiquitin–proteasome pathway has emerged as an important strategy for anti-tumor therapy, and both clinical and preclinical studies have shown the effectiveness of proteasome inhibitors in treating BC [11,12]. Although most drugs target E3 ubiquitin ligases, it is worth exploring the potential usefulness of targeting other ubiquitination-related enzymes, such as Ubiquitin-Activating Enzyme E1 (UBA1).

UBA1, the predominant human E1 enzyme, is highly conserved in eukaryotes [13]. It has been reported that UBA1 enzymes are more actively utilized in several cancer cells, and targeting UBA1 has shown promise as an effective anticancer strategy [14]. Previous studies confirmed that UBA1 might be a promising therapeutic target for BC treatment. However, the precise biological role of UBA1 in BC remains incompletely understood. This study includes a comprehensive analysis to probe into the involvement of UBA1 in BC. Additionally, it aimed to assess its potential as a new prognostic biomarker for individuals with BC.

## 2. Results

### 2.1. UBA1 Is Highly Expressed in the TCGA BC Cohort

Initially, an analysis of *UBA1* in a collection of unpaired samples comprising 33 types of tumors was conducted with the help of the TCGA and GTEx databases. It was observed that the mRNA expression of *UBA1* was significantly heightened in cancer tissues than in their paired paracancerous tissues, particularly among 16 tumor types, including BC (Figure 1A,B). This trend was further validated in paired tumor samples exclusively from the TCGA database (Figure 1C,D), underscoring *UBA1* as a gene with elevated expression in tumor samples.

All individuals with BC in the TCGA database were categorized into two groups as per the median value of *UBA1* expression, and the clinical information for each group is presented in Table 1. Subsequently, a total of 1224 differentially expressed genes (DEGs) were distinguished across the two groups, comprising 68 upregulated and 1156 downregulated genes, as depicted in Figure 1E. The top 15 genes associated with *UBA1* (R > 0.5 and *p* < 0.05) in BC (*KDM5C*, *CDK16*, *ELK1*, *CCDC120*, *FTSJ1*, *ARAF*, *TFE3*, *HDAC6*, *PRICKLE3*, *SMC1A*, *GPKOW*, *NONO*, *HUWE1*, *UBQLN2*, *CCDC22*) are also shown in Figure 1F.

### 2.2. Functional Enrichment Analyses of UBA1-Associated DEGs in BC

Several functional enrichment analyses were performed to ascertain the involvement of *UBA1*-associated DEGs in BC. Immune-related biological processes (BPs), such as immunoglobulin-mediated immune response (GO:0016064), humoral immune response (GO:0006959), immunoglobulin production (GO:0002377), complement activation (GO:0006958), B cell-mediated immunity (GO:0019724), and lymphocyte-mediated immunity (GO:0002449), were found to be enriched (Figure 2A). Similarly, immune-related molecular functions (MFs), including antigen binding (GO:0003823) and immunoglobulin receptor binding (GO:0034987), as well as immune-related cellular components (CCs) like immunoglobulin complex (GO:0019814), were enriched (Figure 2B,C).

In addition to this, the above DEGs were enriched in some REDOX-related MFs, such as aldo-keto reductase (NADP) activity (GO:0004033), oxidoreductase activity (GO:0016616), and alcohol dehydrogenase (NADP+) activity (GO:0008106). Moreover, KEGG analysis (Figure 2D) revealed that UBA1-associated DEGs in BC were involved in xenobiotic metabolism by cytochrome P450 (hsa00980), tyrosine metabolism (hsa00350), chemical carcinogenesis (hsa05204), primary immunodeficiency (hsa05340), drug metabolism-cytochrome P450 (hsa00982), cytokine–cytokine receptor interaction (hsa04060), steroid hormone biosynthesis (hsa00140), nitrogen metabolism (hsa00910), IL-17 signaling pathway (hsa04657), retinol metabolism (hsa00830), and fatty acid degradation (has00071). These findings demonstrate the potential effects of UBA1 on immunity, oxidative stress, and metabolism in BC, which might contribute to disease progression. Additionally, GSEA analysis identified several key signaling pathways potentially associated with *UBA1* in BC, as depicted in Figure 3A–F. These pathways include the KRAS signaling pathway, TNF-α signaling via NF-κB, xenobiotic metabolism, fatty acid metabolism, and IL-6-JAK-STAT3 signaling pathway.

### 2.3. Link Between UBA1 Expression and Infiltrating Immune Cells in BC

A significant positive correlation was observed between *UBA1* expression and Th2 cells (*p* < 0.01), Tgd cells (*p* < 0.05), and Tcm (*p* < 0.05) cells, whereas a significant inverse relationship was reported with ten other immune cell subsets, including Th1 cells, Mast cells, iDC cells, B cells, Tem cells, DC cells, T cells, Cytotoxic cells, CD8T cells, and pDC cells (Figure 4A). Moreover, the cell infiltration levels of B cells, CD8T cells, Cytotoxic cells, DC cells, iDC cells, Mast cells, pDC cells, T cells, Tcm cells, and Th1 cells in the *UBA1* high-expression group significantly differed from those in the low-expression group (*p* < 0.01) (Figure 4B). These findings indicate that *UBA1* might contribute to inhibiting immune responses in BC, thereby potentially contributing to disease development.

### 2.4. Associations Between UBA1 Expression and Clinicopathologic Characteristics in BC

Univariate logistic regression analysis revealed that four clinical characteristics were significantly correlated with *UBA1* expression in BC (Table 2): T stage (OR = 1.540 (1.107–2.142), *p* = 0.010), histological type (OR = 0.553 (0.402–0.760), *p* <0.001), PR status (OR = 1.353 (1.041–1.758), *p* = 0.024), ER status (OR = 1.369 (1.022–1.833), *p* = 0.035). *UBA1* expression was distinctly different in the T stage and histological-type subgroups (Figure 5A,B). Additionally, the relationships between *UBA1* mRNA levels and prognosis were evaluated in these subgroups. The OS results demonstrated that individuals with high *UBA1* expression had notably unfavorable prognoses in several subgroups, such as T1 and T2 (*p* = 0.003), ductal carcinoma (*p* = 0.040), lobular carcinoma (*p* = 0.002), progesterone receptor (PR)-positive (*p* = 0.001), estrogen receptor (ER)-positive (*p* < 0.001) (Figure 5C,D).

### 2.5. Prognostic Values of UBA1 in BC

Univariate and multivariate cox regression analyses aided in demonstrating the association of the *UBA1* mRNA level with clinical indicators (Table 3). To ascertain the prognostic significance of *UBA1* in BC patients, Kaplan–Meier survival and Receiver Operating Characteristic (ROC) curves were generated. As shown in Figure 6A–C (grouping by optimal cutoff value of *UBA1* expression), high *UBA1* expression was linked to poor overall survival (OS, *p* < 0.001), disease-specific survival (DSS, *p* = 0.016), and progress-free survival (PFS, *p* = 0.015) in BC patients. The ROC curve indicated that the *UBA1* expression level had a favorable predictive power (AUC = 0.838, CI:0.810–0.866, Figure 6D). Based on these results, a new nomogram integrating the *UBA1* mRNA level and clinical factors was developed to predict the prognosis of individuals with BC (Figure 6E). The nomogram displayed the independent factors of BC, and the survival rates for 1-, 5-, and 10-year periods anticipated by the nomogram are shown in the line chart (Figure 6F–H). Although DSS (*p* = 0.076) and PFS (*p* = 0.189) were not significant between high- (top 25%) and low- (bottom 25%) expression *UBA1* groups, OS (*p* = 0.006) was significant between the above two groups, which also suggested that *UBA1* expression was associated with OS in breast cancer patients (Appendix A).

### 2.6. Oncogenic Functions of UBA1 in BC

In the results of immunohistochemistry from The Hunam Protein Atlas, it was found that the expression of UBA1 in breast cancer was higher than that in controls (Figure 7A). The oncogenic role of UBA1 in BC was investigated using the MCF-7 and MDA-MB-231 cell lines. *UBA1* was silenced using siRNA, and the knockdown efficiency was validated by RT-qPCR and Western blot (Figure 7B). A CCK8 cell proliferation experiment and colony formation assay established that UBA1 knockdown led to decreased proliferation in BC cells (Figure 7B and Figure 8A). Furthermore, the transwell cell migration experiment showed reduced invasive ability in BC cells after UBA1 silencing (Figure 8B). These findings suggest that UBA1 may promote proliferation and invasion in BC. Additionally, it was found that the expression of *TNF-α* and *IL-6* were both up-regulated in MCF-7 cells when UBA1 was silenced (Figure 9).

## 3. Discussion

BC is a highly heterogeneous disease and remains one of the most lethal malignancies affecting women worldwide [15,16,17,18]. Advancements in molecular characterization and targeted therapies for specific BC subgroups have been ongoing [19]. However, due to the considerable inter- and intra-tumoral heterogeneity of BC, the existing pathological indicators offer limited predictive value for prognosis [20,21]. Hence, new molecular biomarkers must be identified and developed to accurately predict BC prognosis.

UBA1 is a ubiquitinating protein present in both the cytoplasm and nucleus, with consistent protein levels throughout the cell cycle of mammalian cells [22,23]. UBA1 plays a role in regulating cell responses to DNA damage, NF-κB activity, and p53 stability, implying a potential link between UBA1 and tumorigenesis [24,25,26]. Previous studies have associated abnormal UBA1 expression with the progression of hepatocellular carcinoma (HCC), small-cell lung cancer (SCLC), and cutaneous squamous cell carcinoma (SCC) [27,28,29]. Additionally, UBA1 has been recognized as a potential drug target for acute myeloid leukemia and multiple myeloma [30,31]. Despite these findings, the biological role of UBA1 in BC is yet to be comprehended.

The current study explored the involvement of UBA1 in BC. It was observed that both UBA1 mRNA and protein expression were significantly heightened in BC tumor tissues in comparison with adjacent normal tissues in the TCGA database and The Human Protein Atlas. Correlation analysis revealed a close relationship between *UBA1* expression, the T stage, and histological type of BC patients. Traditionally, histological classification of BC mainly depends on estrogen receptor (*ER*), progesterone receptor (*PR*), and human epidermal growth factor receptor 2 (*HER2*) expression, with molecular subtyping improving patient prognosis through hormone therapy [32]. However, the current analysis indicated that among ER- and PR-positive patients, those with high *UBA1* expression had a worse prognosis, while no significant variations were detected in *UBA1* expression among ER- and PR-negative patients. These results suggest a potential association between *UBA1* and ER/PR status.

While BC was historically believed to be poorly immunogenic, recent studies have demonstrated that immunotherapy can activate immune responses, specifically targeting BC tumor cells and improving patient prognosis [33,34]. In this study, the correlation between *UBA1* mRNA levels and tumor-infiltrating immune cells was assessed in individuals with BC by means of ssGSEA. The results revealed that *UBA1* expression was negatively linked to the levels of infiltrating Th1 cells, Mast cells, iDC cells, B cells, Tem cells, DC cells, T cells, Cytotoxic cells, CD8T cells, and pDC cells, and positively correlated with levels of infiltrating Th2 cells (*p* < 0.01), Tgd cells (*p* < 0.05), and Tcm (*p* < 0.05) cells. These outcomes support that UBA1 may contribute to promoting the malignant biological behavior of BC by regulating immune infiltration. Although it was found that the correlation factors between *UBA1* expression and these immune cells mentioned above seemed to be rather low, which might be caused by the poor immunogenicity of BC, we still cannot ignore it because immunotherapy has proven its reliability in tumor treatment. Furthermore, a new nomogram was developed based on *UBA1* expression and clinical characteristics to anticipate the prognosis of individuals with BC, demonstrating its effectiveness and accuracy.

In our functional enrichment analysis, we observed that differentially expressed genes associated with *UBA1* are predominantly involved in immune-related pathways. This alignment with the existing literature reinforces the established notion that *UBA1* plays a pivotal role in immune responses. Moreover, the specificity of these gene expressions to immune pathways suggests a clear direction for our future investigations into the role of *UBA1* in immune function. Considering the complex biological functions of BC, it has been reported that its progression is influenced by TNF-α regulation [35], IL-17 regulation [36], IL-6 regulation [37], oxidative phosphorylation metabolism [38], fatty acid metabolism [39], and immune regulation [40,41]. The functional enrichment analysis depicted that UBA1 contributes to the regulation of the TNF-α signaling pathway, IL-17 signaling pathway, IL-6-JAK-STAT3 signaling pathway, and several immune-related biological processes, highlighting the potential association between UBA1 and BC. Previous studies also showed that TNF-α exhibited pro-apoptotic and anti-mitogenic activities in MCF-7 cells [35], and IL-6 was highly expressed in MCF-7 cells and inhibited the proliferation of MCF-7 cells by inducing apoptosis [37]. Finally, cell-based experiments were carried out to verify the effect of UBA1 on BC and found that UBA1 could impact the expression of *TNF-α* and *IL-6*, acting as an oncogene, promoting cell proliferation and invasiveness in BC.

Ubiquitination is an important biological process, and its stability is closely related to the occurrence and development of diseases. Studies have shown that E3 ubiquitin ligase can affect the expression of IL-6 and TNF-α, then changing their mediated downstream signaling pathway. For example, the membrane-associated E3 ubiquitin ligase MARCH3 could negatively regulate the activation of the IL-6-STAT3 pathway [42]. In addition, silencing the E3 ligase FBXW7 promoted the expression levels of TNF-α, aiding in immune escape [43]. UBA1 is one of the initiating factors of ubiquitination, and changing its expression would affect the ubiquitination process in cells, which might affect the expression of *IL-6* and *TNF-α*. Meanwhile, the expression of *IL-6* and *TNF-α* is also regulated by transcription factors, and these transcription factors are also affected by ubiquitination. For example, *TNF-α* was regulated by the transcription factor *ETV4*, while *ETV4* was also affected by ubiquitination [44,45]. Therefore, although we only detected the mRNA level of *IL-6* and *TNF-α* after silencing UBA1, silencing UBA1 might indirectly affect their mRNA levels by influencing ubiquitination of transcription factors that regulate *IL-6* and *TNF-α*. However, these results remain to be verified by further experiments.

Although this study offers valuable insights, certain limitations should be acknowledged, such as potential data bias in the analysis of single datasets. Further validation using external data is necessary to ensure the robustness of the results. Moreover, due to limitations, we were not able to obtain enough clinical samples of breast cancer to detect the UBA1 protein levels. Data from The Human Protein Atlas alone are not enough to understand the function of UBA1 protein in breast cancer, as the protein is the agent of function. At the same time, lacking normal control cells such as MCF-10A during cell experiments is also one of the deficiencies. We hope that we can supplement these data in the future. In addition, the specific mechanism by which UBA1 regulates tumor immunity in BC must be explored in further in vitro and in vivo studies.

## 4. Materials and Methods

### 4.1. Data Collection and Processing

The clinical data and profiles of mRNA expression for individuals with BC were obtained from the Cancer Genome Atlas (TCGA) database (https://portal.gdc.cancer.gov/projects/TCGA-BRCA (accessed on 10 November 2022)). The Genotype-Tissue Expression Project (GTEx) database provided the mRNA sequencing data of many kinds of normal tissues, commonly used to compare the differential expressions of a specific gene across different cancer types, in conjunction with data from the TCGA database. The UCSC Xena database (https://xenabrowser.net/datapages/ (accessed on 10 November 2022)) provided the combination of the TCGA and GTEx database for detecting the different expression of a gene in different cancer types. Following this, all data were standardized as transcripts per million reads (TPM) prior to analysis. The data from TCGA database were used to perform the major analysis, while data from UCSC database were used to compare the different expression of *UBA1* in different kinds of cancers. Additionally, the results of immunohistochemistry of UBA1 in breast cancer were from The Human Protein Atlas.

### 4.2. Differentially Expressed Genes (DEGs) and Functional Enrichment Analysis

The median value of *UBA1* expression aided in the classification of individuals with BC into low and high *UBA1* expression groups. The R package called “DESeq2” (version 1.36.0) was employed to distinguish DEGs across the two groups. DEGs were selected as per the criteria below: adjusted *p*-value < 0.05 and |log2 fold change (FC)| ≥ 1. The correlation between the expression of DEGs and *UBA1* was assessed by means of Spearman’s correlation analysis. Then, the Kyoto Encyclopedia of Genes and Genomes (KEGG), Gene Ontology (GO), and Gene Set Enrichment Analysis (GSEA) were carried out on selected DEGs with the help of the R package called “clusterProfiler” (version 4.4.4) [46].

### 4.3. Immune Infiltration Analysis

Single-sample gene set enrichment analysis (ssGSEA) helped calculate the relative enrichment scores of 24 immune cells, as per specific sets of genes [47]. To ascertain the association of *UBA1* expression with immune infiltration in BC [48], Spearman’s correlation analysis was conducted. Additionally, the differential levels of immune infiltration across the two *UBA1* expression groups were evaluated by means of the Wilcoxon rank-sum test.

### 4.4. Association of UBA1 Expression with Clinical Information in Individuals with BC

Univariate logistic regression analysis was applied to ascertain the relationship between *UBA1* expression groups (low and high expression) and other clinical subgroups in BC patients. Subgroups with *p* < 0.05 were considered significant. Furthermore, the influence of *UBA1* expression on patient prognosis in these subgroups was explored using survival plots with the log-rank test. To avoid increasing the risk for a type I statistical error, our grouping method in log-rank test included grouping by optimal cutoff value of *UBA1* expression or grouping by high- (top 25%) and low- (bottom 25%) expression *UBA1*.

### 4.5. Development and Verification of the Nomogram

A nomogram was developed to anticipate OS probability with the aid of independent prognostic factors obtained via univariate and multivariate Cox regression analyses. Furthermore, calibration plots aided in the assessment of the efficacy of the nomogram, whereas the concordance index (C-index) helped quantify its discrimination. Significant factors were determined with *p* < 0.1 in the univariate Cox regression analysis, while factors with *p* < 0.05 were deemed significant in the multivariate Cox regression analysis. R package “survival [3.3.1]” and “rms [6.3-0]” were used to develop a nomogram.

### 4.6. Cell Culture and RNA Interference

Cell lines of BC, MCF-7 and MDA-MB-231, were retrieved from the American Type Culture Collection (ATCC, Manassas, VA, USA) and cultured in DMEM containing 10% fetal bovine serum (FBS) and 1% penicillin-streptomycin (Gibco, Carlsbad, CA, USA) in a humidified atmosphere with 5% CO_2_ at 37 °C. The siRNA targeting UBA1 was transfected with the help of Lipofectamine 2000 Reagent (Thermo Fisher Scientific, Waltham, MA, USA). The RNA oligo sequences for UBA1 were as follows: siUBA1-1: GTGCTATGGTTTCTATGGTTA, siUBA1-2: CCACTGCCTTCTACCTTGTTT.

### 4.7. Quantitative Reverse-Transcription Polymerase Chain Reaction (RT-qPCR)

The entire RT-qPCR process, including reverse transcription, the extraction of total RNA, and qPCR techniques, was executed using a commercial kit (Accurate Biology, Shanghai, China, Cat. No: AG21023, AG11706, and AG11718). The primer pair of UBA1, TNF-α and IL-6 was obtained from the ORIGENE (https://www.origene.com.cn/ (accessed on 27 March 2023)) and had the given sequences. UBA1-F: 5′-TCCTCACAGAGGACAAGTGCCT-3′, UBA1-R: 5′-CTTGAGCAGCTCACAGCCAATG-3′; TNF-α-F: 5′-CTCTTCTGCCTGCTGCACTTTG-3′, TNF-α-R: 5′-ATGGGCTACAGGCTTGTCACTC-3′; IL-6-F: 5′-AGACAGCCACTCACCTCTTCAG-3′, IL-6-R: 5′-TTCTGCCAGTGCCTCTTTGCTG-3′.

### 4.8. Western Blot

Total protein was extracted by using commercial kit (Thermo Fisher Scientific; 78501), and protein quantification was examined using BCA method (Thermo Fisher Scientific; 23227) before the Western blot (WB). These two procedures were performed according to the kit’s instructions.

The extracted protein was uniformly loaded onto the 7.5% SDS-PAGE gel (Shanghai Epizyme Biomedical Technology Co., Ltd., Shanghai, China; PG111) and then transferred to the 0.45 µm nitrocellulose membrane (Millipore, HATF00010, Bellerica, MA, USA). The membrane was blocked with 5% non-fat milk at room temperature about 1 h, and then incubated with primary antibodies like anti-UBA1 (Abcam, ab181225, 1:1000, Cambridge, MA, USA) and anti-β-Actin (ABclonal, AC026, 1:10,000,Wuhan, China) at 4 °C overnight. Thereafter, the membrane was washed with Tris Buffered Saline with Tween-20 (TBST) three times (10 min/time) at room temperature. Then, the membrane was incubated with HRP-conjugated Goat anti-Rabbit IgG (H + L) secondary antibodies (ABclonal, AS014, 1:5000) at room temperature about 2 h and then washed 3 times for 10 min each with TBST. Finally, the membrane was detected by the electrochemiluminescence (ECL) imaging using the ChemiDoc XRS+ System (Bio-Rad, Laboratories, Hercules, CA, USA).

### 4.9. CCK-8 Cell Proliferation Assay and Colony Formation Assay

Following digestion, counting, and centrifugation, cells transfected with siUBA1 (including a negative control group) were seeded into 96-well plates at a density of 3000 cells/well. This was followed by culturing. Cell Counting Kit-8 (CCK-8) reagent (biosharp, Hefei, China) was added to the cells (10 μL per well) and incubated for 2 h. Absorbance at 450 nm was recorded at varying time points (0, 1, 3, 5 days). Next, the cells were seeded into a 6-well plate (MCF-7: 1000 cells/well; MDA-MB-231: 500 cells/well) for the colony formation assay and cultured for one week. Additionally, the cells were fixed with 4% paraformaldehyde (*v*/*v*) and stained with 0.1% crystal violet (*v*/*v*).

### 4.10. Cell Migration Assay

Transwell chambers having a pore size of 8 mm (Corning, NY, USA) were employed for the migration assay. Cells were then resuspended in 200 μL of serum-free medium and placed inside the upper chamber. The lower chambers were filled with 500 μL of DMEM containing 10% FBS and subjected to incubation. The number of cells in the migration assay was 1 × 10^5^ for both types of cell lines, while the incubation time for two types of cell lines was different (MCF7: 48 h; MDA-MB-231: 10 h). The cells were fixed with 4% paraformaldehyde (*v*/*v*) and then stained with 0.1% crystal violet (*v*/*v*). The migrating cells were counted with the help of Image-Pro Plus 6.0 software, and bright-field images were captured with the aid of an Olympus inverted microscope (Media Cybernetics, Bethesda, MD, USA).

### 4.11. Detecting TNF-α and IL-6

As TNF-α and IL-6 are the initiators of TNF-α signaling and IL-6-JAK-STAT3 pathways, we detected the mRNA expression of *TNF-α* and *IL-6* after treating E1 Ubiquitin-Activating Enzyme inhibitor TAK-243 (MedChemExpress, Junction, NJ, USA, #HY-100487) in MCF-7 cells (0.5 μM, 12 h).

### 4.12. Statistical Analysis

The quantitative and qualitative clinical data were presented as median and counts (or percentage) values, respectively. The bioinformatic analyses were performed by R software (version 4.2.1). Log-rank test was used to analyze the survival outcomes, and Wilcoxon test was employed to find significant DEGs, GO enrichment, GSEA and immune infiltration. As for biological assays, ImageJ (2022) was employed to detect the count of colony formation and migration assays. The statistical analyses of all assays were performed by *t* test or two-way ANOVA using GraphPad Prism 8.0, and *p* < 0.05 was regarded as statistically significant. All assays were repeated 3 times and showed with representative findings in the manuscript.

## 5. Conclusions

This study elucidates the oncogenic function of UBA1 in BC and suggests a potential link between UBA1 and tumor–immune interactions. Furthermore, the utility of UBA1 as a new prognostic biomarker was established for anticipating the prognostic outcomes of individuals with BC. These findings underscore the importance of developing and applying therapeutic targets for UBA1 to facilitate clinical treatment options for BC.

## Figures and Tables

**Figure 1 ijms-25-12696-f001:**
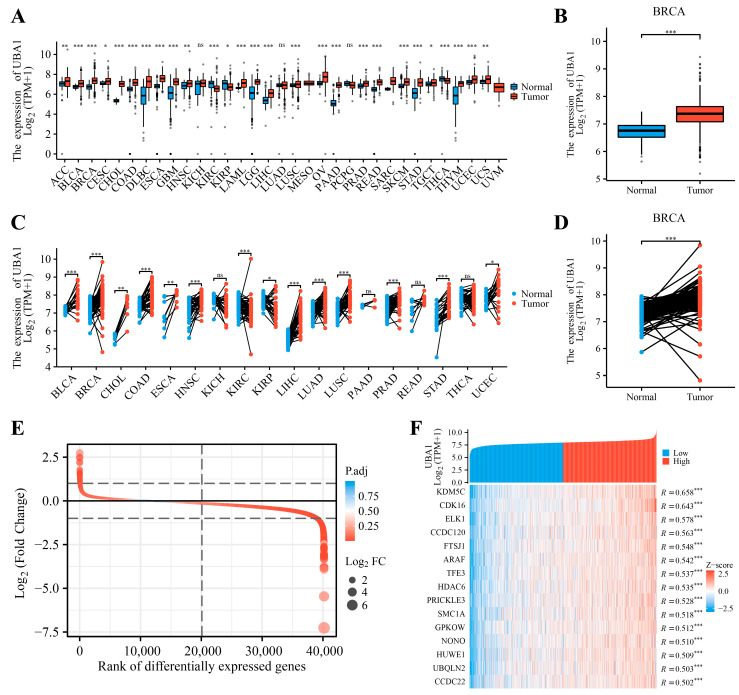
*UBA1* mRNA expression levels in various types of human tumors. (**A**) *UBA1* mRNA expression levels in unpaired samples of 33 types of tumors relative to healthy tissues from TCGA and GTEx database. (**B**) Expression of *UBA1* mRNA in BC vs. unpaired normal tissues in TCGA and GTEx database. (**C**) *UBA1* mRNA expression levels in 18 types of tumors compared with paired paracancerous tissues from the TCGA database. (**D**) *UBA1* mRNA expression levels in BC and paired paracancerous in TCGA database. (**E**) *UBA1*-related DEGs. (**F**) Heatmap of association between *UBA1* expression and *UBA1*-related DEGs (R > 0.5). ns, *p* > 0.05,* *p* < 0.05, ** *p* < 0.01, *** *p* < 0.001.

**Figure 2 ijms-25-12696-f002:**
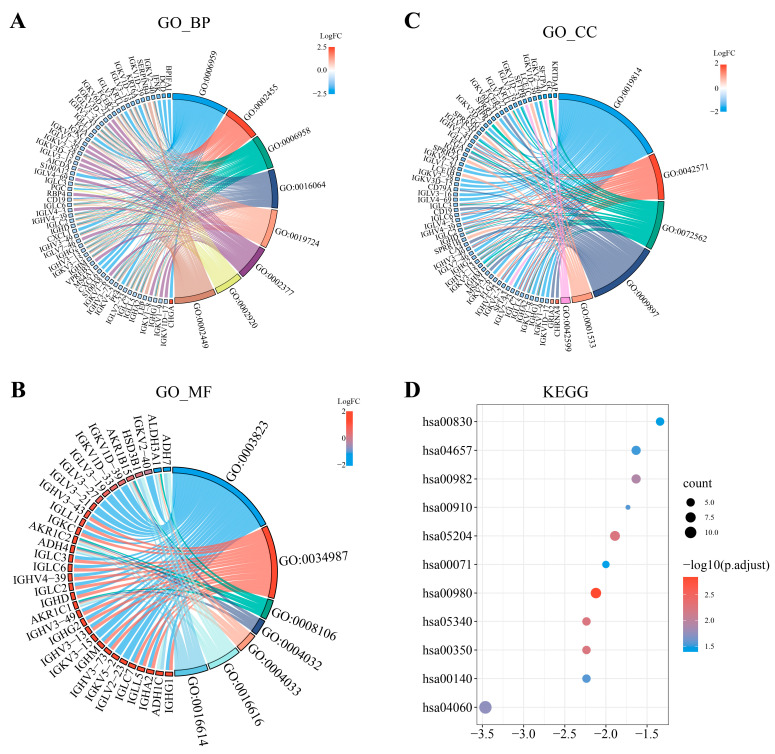
GO enrichment analysis and KEGG pathway analysis of *UBA1*-related DEGs in BC. (**A**–**C**) GO analysis of UBA1-linked DEGs in BC. BP, biological process; CC, cellular component; MF, molecular function. The colors of different lines correspond to different GO Terms. (**D**) Dot plot visualizing the results of KEGG analysis of UBA1-linked DEGs in BC.

**Figure 3 ijms-25-12696-f003:**
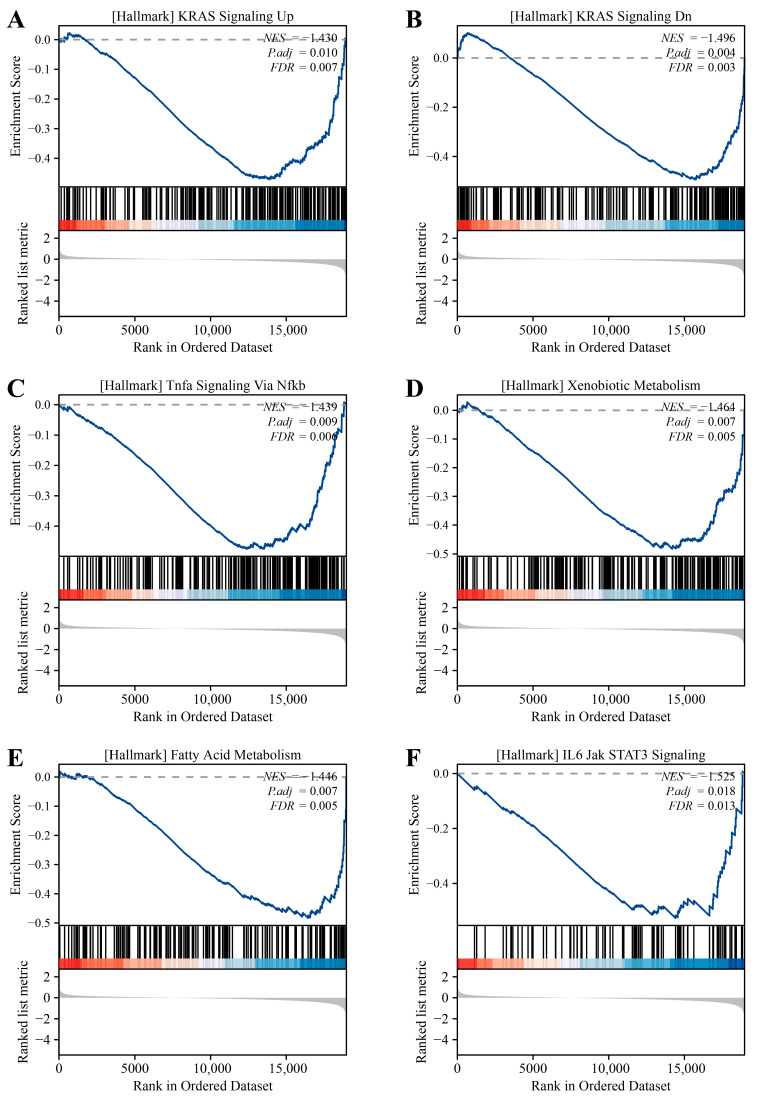
GSEA analysis of *UBA1*-related DEGs in BC. (**A**–**F**) GSEA analysis identified six key signaling pathways potentially associated with *UBA1* in BC. The GSEA enrichment score is represented by the blue curve.

**Figure 4 ijms-25-12696-f004:**
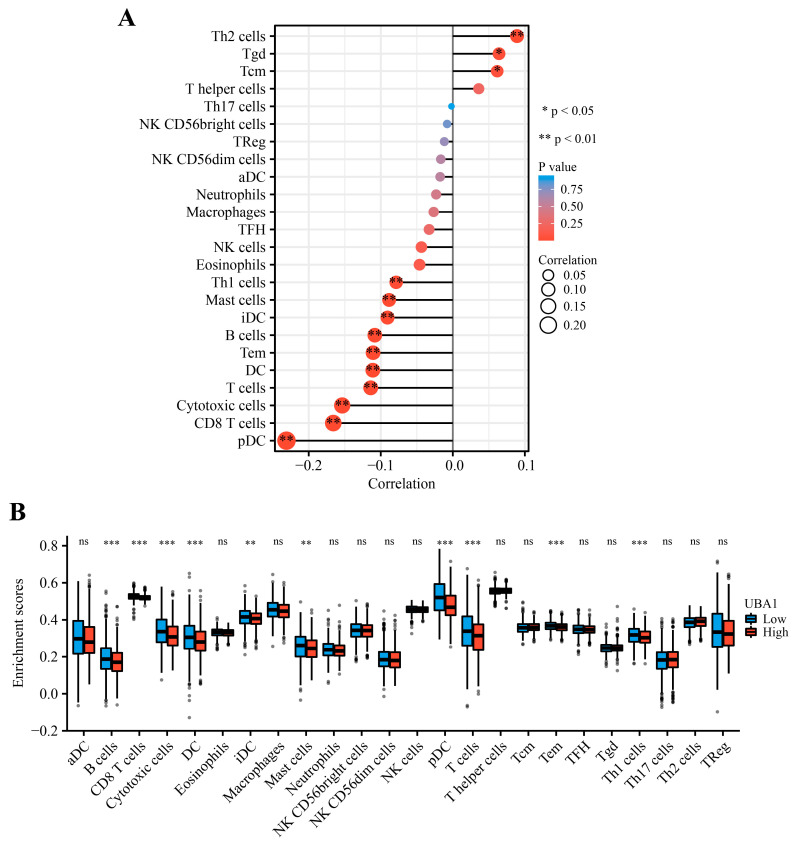
*UBA1* mRNA expression level correlates with immune infiltration in BC. (**A**) Association of *UBA1* expression with the relative abundance of 24 types of immune cells in BC. (**B**) Comparison of immune infiltration of 24 types of immune cells across *UBA1* high- and low-expression groups. Box plots represent the median, the spread of the middle 50% of the data (interquartile range), and the 10th and 90th percentiles, with individual dots signifying studies that are considered outliers. ns, *p* > 0.05, ** *p* < 0.01, *** *p* < 0.001.

**Figure 5 ijms-25-12696-f005:**
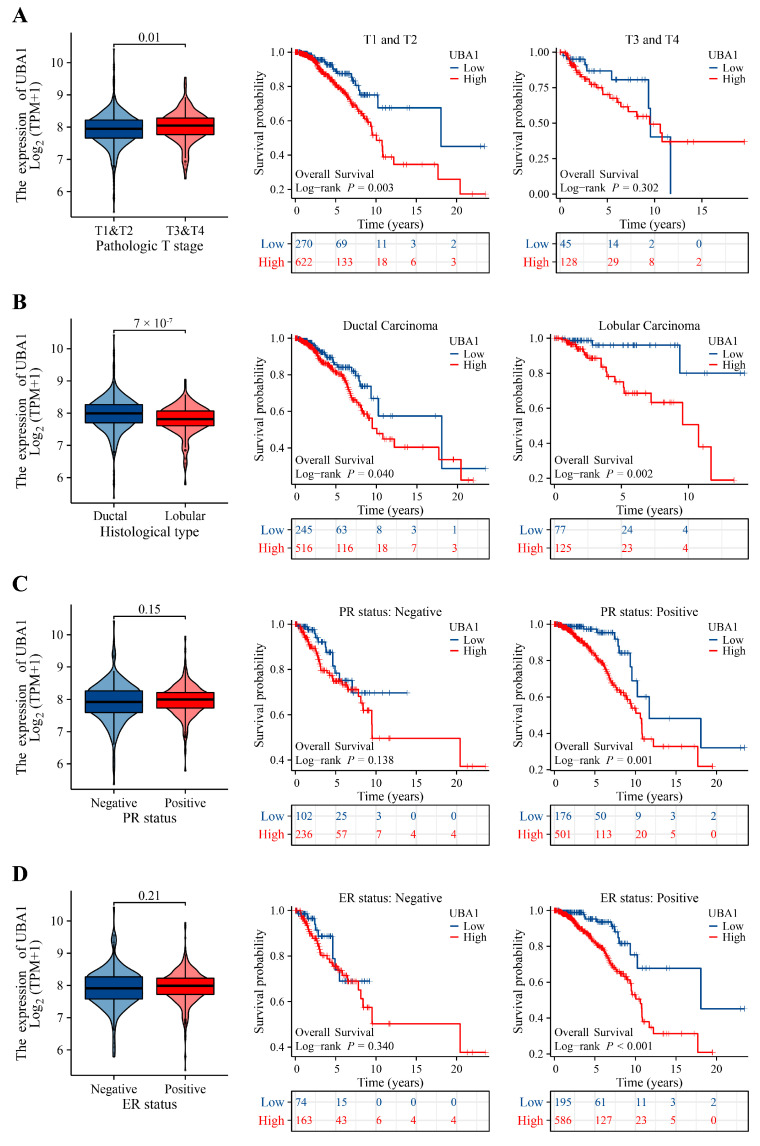
Associations between *UBA1* expression and clinicopathological factors in BC. Expression levels and prognostic values of *UBA1* in (**A**) T stage; (**B**) Histological type; (**C**) PR status; and (**D**) ER status. ER, estrogen receptor; PR, progesterone receptor.

**Figure 6 ijms-25-12696-f006:**
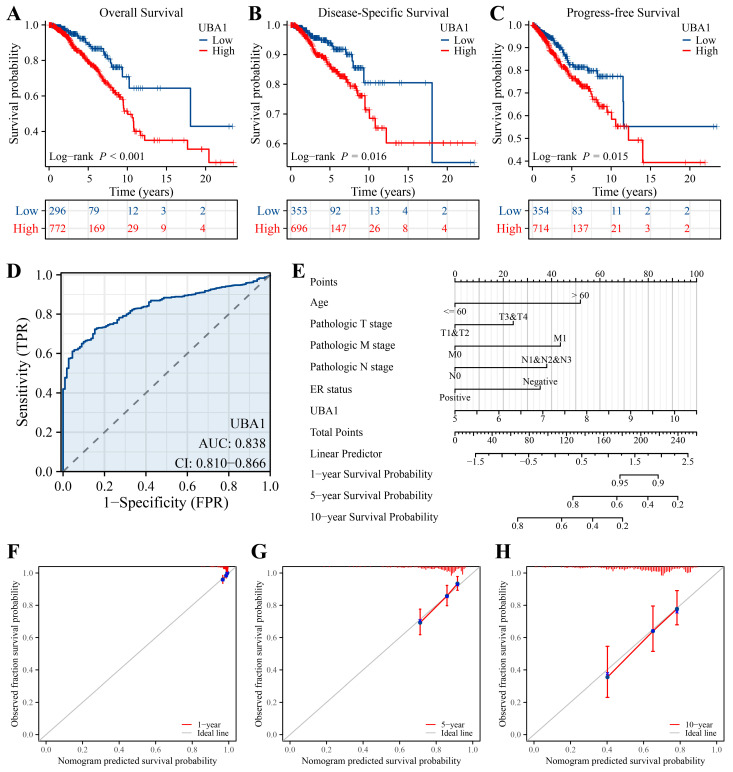
Prognostic values of *UBA1* in BC. Kaplan–Meier survival curve showing the prognostic values of *UBA1* expression in individuals with BC: (**A**) overall survival; (**B**) disease-specific survival; and (**C**) progress-free survival. (**D**) ROC curves showing the potential predictive power of *UBA1* in BC. The area under the ROC curve was used to evaluate the predictive ability of the model. (**E**) Nomogram showing the correlation between the *UBA1* expression level and clinical indicators. (**F**–**H**) Calibration of the nomogram of 1-, 5- and 10-year overall survival rate in BC patients.

**Figure 7 ijms-25-12696-f007:**
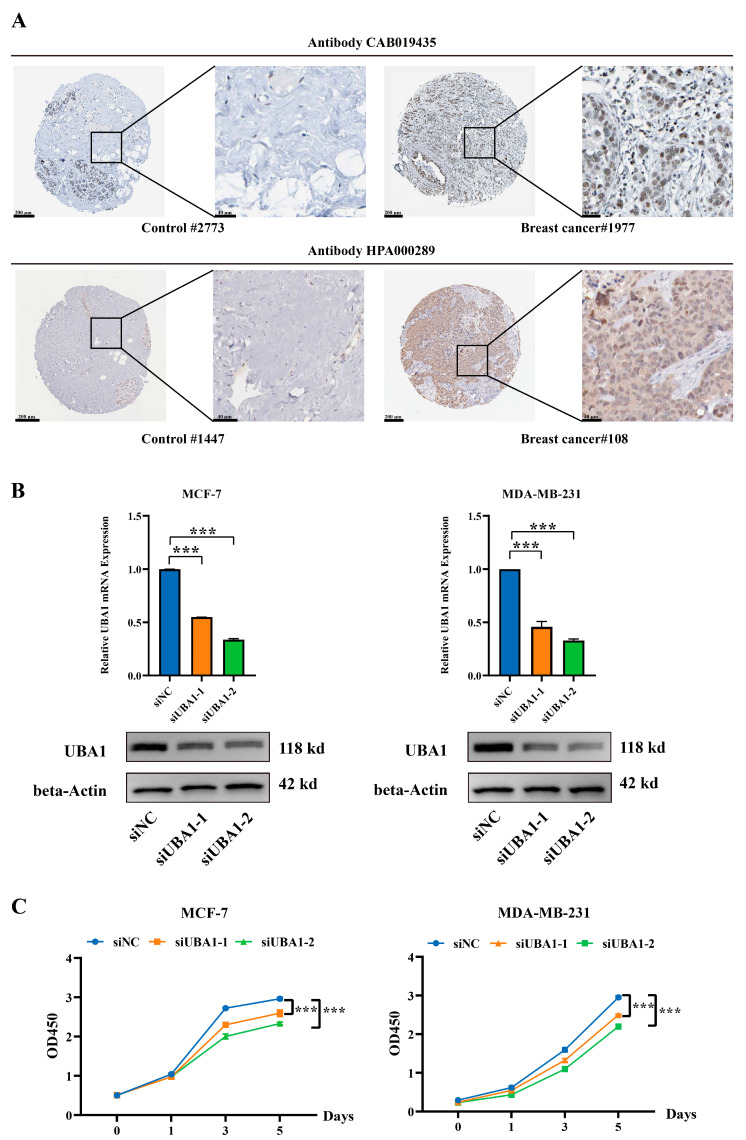
Silencing UBA1 inhibit the growth of MCF-7 and MDA-MB-231 cells. (**A**) The different expression of UBA1 of immunohistochemistry in breast cancer and controls from The Human Protein Atlas. (**B**) Assessment of silencing efficacy of UBA1 by RT-qPCR (*t* test) and Western Blot. (**C**) CCK-8 assay to measure cell proliferation (two-way ANOVA). *** *p* < 0.001.

**Figure 8 ijms-25-12696-f008:**
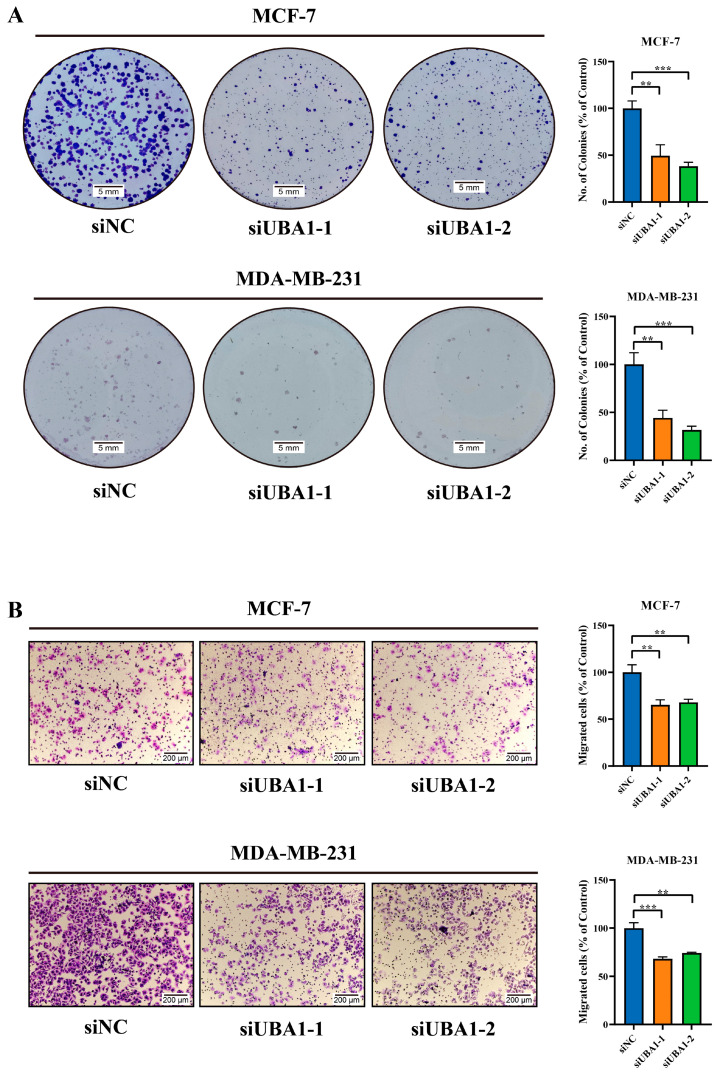
Silencing UBA1 inhibits the colony and migration ability of MCF-7 and MDA-MB-231 cells. (**A**) Colony formation assay to investigate cell growth ability. (**B**) Detection of cell migration using a migration assay (scale bar: 100 µm, *t* test). ** *p* < 0.01, *** *p* < 0.001.

**Figure 9 ijms-25-12696-f009:**
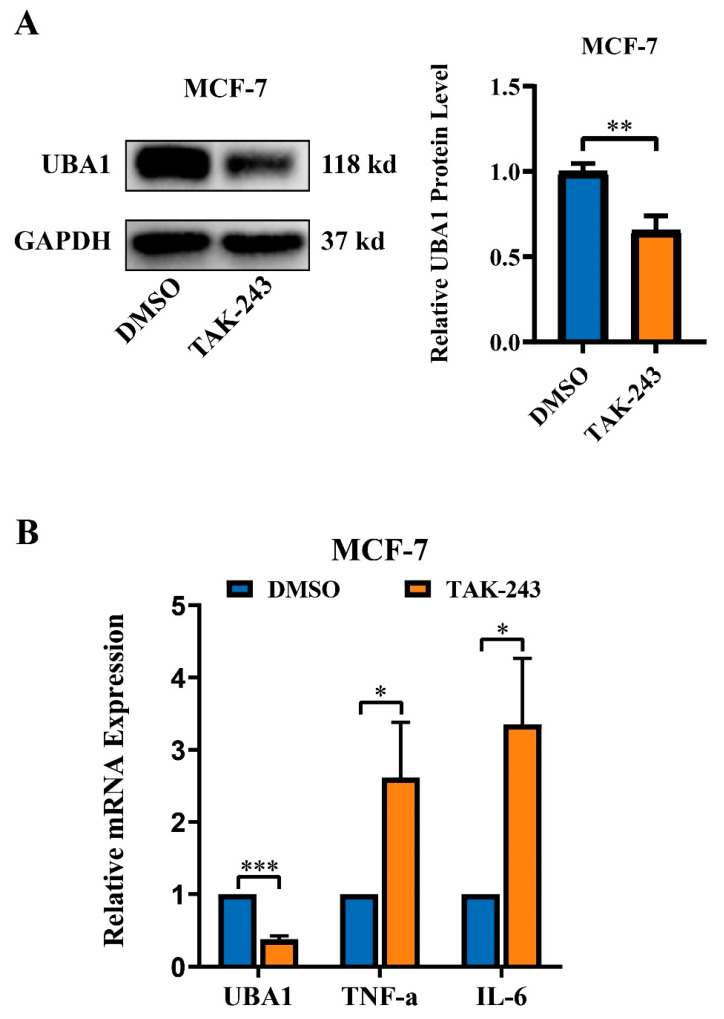
Silencing UBA1 increases the mRNA expression of *TNF-α* and *IL-6* in MCF-7 cells. (**A**) Assessment of the protein level of UBA1 after treating TAK-243 in MCF-7 cells (0.5 μM, 12 h). (**B**) Assessment of the mRNA expression of *TNF-α* and *IL-6* by RT-qPCR (*t* test). * *p* < 0.05,** *p* < 0.01, *** *p* < 0.001.

**Table 1 ijms-25-12696-t001:** The clinical characteristics of BC patients.

Characteristics	Low Expression of UBA1 (n = 534)	High Expression of UBA1 (n = 535)	*p* Value
Age, median (IQR)	59 (49, 67)	58 (48, 67.5)	0.551
Age, n (%)			0.647
≤60	291 (27.2%)	299 (28%)	
>60	243 (22.7%)	236 (22.1%)	
Race, n (%)			0.465
Asian	26 (2.7%)	34 (3.5%)	
Black or African American	93 (9.5%)	87 (8.9%)	
White	381 (38.9%)	358 (36.6%)	
T stage, n (%)			0.036
T1	150 (14.1%)	126 (11.8%)	
T2	312 (29.3%)	305 (28.6%)	
T3	59 (5.5%)	79 (7.4%)	
T4	12 (1.1%)	23 (2.2%)	
M stage, n (%)			0.231
M0	433 (47.5%)	459 (50.3%)	
M1	7 (0.8%)	13 (1.4%)	
N stage, n (%)			0.390
N0	262 (25%)	247 (23.5%)	
N1	162 (15.4%)	188 (17.9%)	
N2	58 (5.5%)	58 (5.5%)	
N3	41 (3.9%)	34 (3.2%)	
Pathologic stage, n (%)			0.193
Stage I	101 (9.7%)	80 (7.7%)	
Stage II	301 (28.8%)	305 (29.2%)	
Stage III	116 (11.1%)	124 (11.9%)	
Stage IV	6 (0.6%)	12 (1.1%)	
Histological type, n (%)			<0.001
Infiltrating Ductal Carcinoma	364 (37.8%)	397 (41.2%)	
Infiltrating Lobular Carcinoma	126 (13.1%)	76 (7.9%)	
PR status, n (%)			0.048
Negative	184 (18%)	154 (15.1%)	
Positive	318 (31.2%)	360 (35.3%)	
Indeterminate	1 (0.1%)	3 (0.3%)	
ER status, n (%)			0.107
Negative	131 (12.8%)	106 (10.4%)	
Positive	371 (36.3%)	411 (40.3%)	
Indeterminate	1 (0.1%)	1 (0.1%)	
HER2 status, n (%)			0.454
Negative	266 (37%)	284 (39.5%)	
Positive	77 (10.7%)	80 (11.1%)	
Indeterminate	8 (1.1%)	4 (0.6%)	
PAM50, n (%)			0.539
Lum A	268 (26%)	285 (27.7%)	
Lum B	92 (8.9%)	112 (10.9%)	
Her2	42 (4.1%)	40 (3.9%)	
Basal	99 (9.6%)	91 (8.8%)	
Menopause status, n (%)			0.271
Pre	105 (10.9%)	120 (12.5%)	
Peri	22 (2.3%)	17 (1.8%)	
Post	364 (37.9%)	332 (34.6%)	
Anatomic neoplasm subdivisions, n (%)			0.562
Left	273 (25.5%)	283 (26.5%)	
Right	261 (24.4%)	252 (23.6%)	

**Table 2 ijms-25-12696-t002:** The results of univariate logistic regression analysis.

Characteristics	Total (N)	Odds Ratio (OR)	*p* Value
Age (>60 vs. ≤60)	1069	0.945 (0.743–1.203)	0.647
Race (White vs. Non-white)	979	0.924 (0.691–1.237)	0.595
T stage (T3&T4 vs. T1&T2)	1066	1.540 (1.107–2.142)	**0.010**
M stage (M1 vs. M0)	912	1.752 (0.692–4.432)	0.236
N stage (N1&N2&N3 vs. N0)	1050	1.138 (0.893–1.450)	0.296
Pathologic stage (Stage III&IV vs. Stage I&II)	1045	1.164 (0.878–1.542)	0.290
Histological type (Lobular vs. Ductal)	963	0.553 (0.402–0.760)	**<0.001**
PR status (Positive vs. Negative)	1016	1.353 (1.041–1.758)	**0.024**
ER status (Positive vs. Negative)	1019	1.369 (1.022–1.833)	**0.035**
HER2 status (Positive vs. Negative)	707	0.973 (0.682–1.388)	0.880
PAM50 (Her2&Basal vs. Lum A&Lum B)	1029	0.842 (0.638–1.112)	0.226
Menopause status (Post&Peri vs. Pre)	960	0.791 (0.587–1.067)	0.125
Anatomic neoplasm subdivisions (Right vs. Left)	1069	0.931 (0.733–1.184)	0.562

Meaningful independent variables are indicated with bold face capital letters.

**Table 3 ijms-25-12696-t003:** The Cox regression analyses in BRAC patients (OS).

Characteristics	N	Univariate Analysis		Multivariate Analysis	
		Hazard Ratio (95% CI)	*p*	Hazard Ratio (95% CI)	*p*
Age					
≤60 years old	590	Reference		Reference	
>60 years old	478	2.040 (1.471–2.828)	**<0.001**	3.254 (1.732–6.113)	**<0.001**
Race					
Non-white	239	Reference			
White	739	0.877 (0.591–1.302)	0.515		
T stage					
T1&T2	892	Reference		Reference	
T3&T4	173	1.653 (1.138–2.399)	**0.008**	2.691 (1.401–5.170)	**0.003**
M stage					
M0	892	Reference		Reference	
M1	20	4.339 (2.515–7.486)	**<0.001**	5.175 (1.681–15.924)	**0.004**
N stage					
N0	509	Reference		Reference	
N1&N2&N3	540	2.138 (1.492–3.064)	**<0.001**	1.908 (1.080–3.371)	**0.026**
Histological type					
Infiltrating Ductal Carcinoma	761	Reference			
Infiltrating Lobular Carcinoma	202	0.864 (0.549–1.360)	0.527		
PR status					
Negative	338	Reference			
Positive	677	0.759 (0.539–1.069)	0.115		
ER status					
Negative	237	Reference		Reference	
Positive	781	0.701 (0.485–1.013)	**0.059**	0.372 (0.209–0.661)	**<0.001**
HER2 status					
Negative	550	Reference		Reference	
Positive	157	1.611 (0.981–2.644)	**0.059**	0.834 (0.432–1.610)	0.589
PAM50					
Lum A&Lum B	756	Reference			
Her2&Basal	272	1.270 (0.888–1.817)	0.190		
Menopause status					
Pre	225	Reference		Reference	
Peri&Post	734	2.096 (1.245–3.529)	**0.005**	2.216 (0.890–5.517)	0.087
Anatomic neoplasm subdivisions					
Left	556	Reference			
Right	512	0.780 (0.562–1.083)	0.139		
UBA1	1068	1.516 (1.102–2.085)	**0.010**	1.917 (1.098–3.347)	**0.022**

Meaningful independent variables are indicated with bold face capital letters.

## Data Availability

Data are contained within the article.

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
