# Peer review of "Ubiquitin-Activating Enzyme E1 (UBA1) as a Prognostic Biomarker and Therapeutic Target in Breast Cancer: Insights into Immune Infiltration and Functional Implications"

_ijms, 2024, doi:10.3390/ijms252312696_

Round 1

Reviewer 1 Report (New Reviewer)

Comments and Suggestions for Authors

Mingtao Feng et al. in “Ubiquitin-Activating Enzyme E1 (UBA1) as a Prognostic Biomarker and Therapeutic Target in Breast Cancer: Insights into Immune Infiltration and Functional Implications” show in extremely depth how the silencing UBA1 significantly impeded the growth and development of BC cell lines.

These findings highlight UBA1 as a potential prognostic biomarker linked to immune infiltration in Breast Cancer.

I consider original the proposal to consider UBA1 as a potential prognostic 20 biomarker linked to immune infiltration in BC.

The references are appropriate and recent. They support the conceptualizations present in the article.

The authors should improve the representative image of Figure 8.A and the 8.B images should show the entire well, not just one part.

Author Response

We provide a point-by-point response to the reviewer’s comments and upload it as a PDF file.

Reviewer 2 Report (New Reviewer)

Comments and Suggestions for Authors

This manuscript describes studies aiming at demonstrating the importance of UBA1 for breast cancer. Public gene expression data were analyzed and siRNA mediated UBA1 knocck down in two breats cancer cell lines performed. The authors conclude that UBA1 is indeed correlated with BC progression and immune cell infiltration. BC cells showed decreased proliferation, invasion and colony formation upon UBA1 reduction. An ubiquitin inhibitor also reduced UBA1 protein and increased IL6 and TNFalpha. Altogether, UBA1 has the potential as prognostic marker and target for BC.

This is an interesting topic and fits to the scope of this journal.

I am not really an expert on bioinformatics analyses; therefore, my comments on this topic can only be superficial.

I have the following points of critic with this manuscript that should be resolved:

The method section seems sometimes too short. Please indicate in more detail, which databases have been analyzed and which source was used for downloading. I do not understand how the nomogram was constructed; Was there any software used?

Western blot detection method is not explained.

Protein data from patients e.g. determined by immunohistochemistry are widely missing. The mentioned data from the human protein atlas usually refer to the TCGA RNA data. The authors should clarify how this conclusion was obtained.

An analysis of ubiquitinylation in BC would be helpful to support the conclusions. This is also relevant for the cell culture experiments.

A siRNA mediated KO in non cancerous cells would greatly strengthen the conclusion that UBA1 could be a treatment target.

A UBA1 silencing resulted in IL6 and TNFalpha increase. Please discuss the possible implementations on BC in more detail.

In fig. 4A, the correlation with immune cells is shown. The correlation factor seems rather low. Is this really relevant? Can you compare this result with other studies?

For the Kaplan Meier curves, how was the cut off determined?

In Table 1 survival states were correlated with UBA1 status. This makes only sense when the follow up time was fixed. Please explain how this correlation was determined. There is also cox regression and Kaplan Meier curves, so these data might be deleted.

Table 2: Does this concern OS? Please improve the legend. Here the N-stage is not significant for survival. In most studies, N is one of the most significant parameter for survival. Please discuss this. You may try comparing N0 + N1 with N2 + N3 or do the analysis without cut-off.

Table 3: Were all the parameters with p < 0.01 (?) included into the multivariable cox regression analysis?

Line 377. What is this reference?

Author Response

We provide a point-by-point response to the reviewer’s comments and upload it as a PDF file.

Round 2

Reviewer 2 Report (New Reviewer)

Comments and Suggestions for Authors

Please refer to the attached file!

Author Response

Dear editors and reviewers,

Thank you very much for giving chance for revision. We made a point-by-point response to editors and reviewers in the order of receiving the comments. Please kindly see the specific responses as follows:

Title of the Manuscript: Ubiquitin-Activating Enzyme E1 (UBA1) as a Prognostic Biomarker and Therapeutic Target in Breast Cancer: Insights into Immune Infiltration and Functional Implications.

Reviewer 2

Thank you very much for giving comments for our manuscript. We made a point-by-point response to your comments. Please see the specific response as follows:

Comment 3:  I am not convinced by the statements on the proteinatlas, as the figure you presented clearly show the expression values as TPM (transcripts per million). In addition, the information panel (see below) refers to the mRNA levels in TPKM. You might use the few examples for protein abundance, but I suggest removing the statements on survival concerning this website.

Nevertheless, you can try downloading MS data from the TCGA breast cancer proteome, although

there are not all proteins covered.

The insufficient number of protein data on UBA1 should be discussed.

Authors’ Response: We are grateful for your meticulous review and the important points you have raised regarding the use of data from The Human Protein Atlas and the discussion of protein data on UBA1. Firstly, we acknowledge your concern about the discrepancy between the TPM values presented in our figure and the information panel referring to mRNA levels in TPKM.  We appreciate your guidance on this matter and have taken immediate action to revise our manuscript.  In line with your suggestion, we have carefully removed the statements related to survival data from The Human Protein Atlas (specifically, lines 194-197 in the original manuscript) to ensure that our presentation is accurate and consistent with the data we have analyzed.

Additionally, we have addressed your suggestion regarding the limited protein data on UBA1.  We have included a discussion on the small sample size of protein data available for UBA1, highlighting the implications of this limitation for our findings and the need for further research with a larger dataset.

We hope that these revisions have adequately addressed your concerns and have strengthened the manuscript.  We are committed to maintaining the highest standards of scientific rigor and thank you for your constructive feedback. Please kindly see the manuscript.

Change to Text: Please kindly see the highlights (blue) in the revised manuscript. Line 258-264.

Comment 5: I understood that you cannot add data using MCF-10 at this time. This is sad and a clear weakness of

this manuscript. Please discuss this point in the discussion.

Authors’ Response: Thank you very much for your valuable feedback.  We understand and appreciate the significance of including data using MCF-10 in our study, and we regret that we are unable to do so at this time.  We acknowledge that this represents a limitation of our current manuscript. In response to your suggestion, we have carefully considered how to address this point in the discussion section of our manuscript.  We have added a detailed discussion on this limitation, explaining the reasons for our inability to include MCF-10 data at this stage and the impact this may have on the interpretation of our results.  We believe that this additional discussion will provide readers with a clearer understanding of the constraints faced during our research and the potential implications for future studies.

We hope that this amendment satisfies your request and addresses the concern raised.  We are grateful for your guidance and are committed to enhancing the quality and transparency of our work.

Change to Text: Please kindly see the highlights (blue) in the revised manuscript. Line 258-264.

Comment 7: From my point of view, you can leave Fig. 4A as long as you discuss the low correlation factors.

Authors’ Response: Thank you very much for your insightful comment regarding Figure 4A.   We appreciate your understanding of the complexities involved in our data analysis and the value you place on retaining this figure in the manuscript. In response to your suggestion, we have taken the time to thoroughly discuss the low correlation factors in the context of Figure 4A.   We have expanded upon the potential reasons for these low correlations and how they may impact the interpretation of our results.   This discussion is now integrated into the relevant section of the manuscript(Line 216-219), where we believe it provides a more nuanced understanding of the data presented in Figure 4A.

We hope that this additional discussion meets with your approval and addresses your concerns. Please find the updated discussion in the manuscript, and we would be delighted to receive any further feedback you may have. Thank you once again for your valuable input.

Change to Text: Please kindly see the highlights (blue) in the revised manuscript. Line 216-219.

Comment 8: By optimizing the cutoff in a survival analysis, you increase the risk for a type I statistical error (false

positive). I know that this is frequently done, however, you should be sure that statistical significance

is present over a wide range of cut-off values.

Authors’ Response: We are grateful for your constructive suggestion and appreciate your concern regarding the potential for type I statistical error in our survival analysis.  Your expertise in this area is highly valued, and we have taken your comments seriously. In response to your feedback, we have re-evaluated our approach to determining the optimal cut-off value for survival analysis.  As you rightly pointed out, this is a widely used method in Kaplan-Meier (KM) curves.   Upon further review of the literature, we identified that some papers have used the top and bottom 25% of expression levels to categorize high and low groups for prognosis analysis (1). Inspired by this method, we conducted an additional analysis to compare the survival states between groups with high (top 25%) and low (bottom 25%) UBA1 expression levels.

Our findings revealed significant differences in overall survival (OS) between these two groups, which aligns with our primary focus on OS throughout the paper, particularly in the Cox regression analysis presented in Table 3.  Although we observed no significant differences in disease-specific survival (DSS) and progression-free survival (PFS), we believe that our comprehensive analysis, with a focus on OS, supports the conclusion that UBA1 is an independent prognostic factor in breast cancer patients.

We have incorporated this additional analysis and the rationale behind our approach into the manuscript to provide a more robust discussion and to address your concerns about the statistical significance across a range of cut-off values.

Thank you once again for your insightful comments.  We hope that our revisions and the inclusion of this additional analysis have adequately addressed your concerns.

(1) Zhao Y, Liu Z, Liu G, Zhang Y, Liu S, Gan D, Chang W, Peng X, Sung ES, Gilbert K, Zhu Y, Wang X, Zeng Z, Baldwin H, Ren G, Weaver J, Huron A, Mayberry T, Wang Q, Wang Y, Diaz-Rubio ME, Su X, Stack MS, Zhang S, Lu X, Sheldon RD, Li J, Zhang C, Wan J, Lu X. Neutrophils resist ferroptosis and promote breast cancer metastasis through aconitate decarboxylase 1. Cell Metab. 2023 Oct 3;35(10):1688-1703.e10. doi: 10.1016/j.cmet.2023.09.004. PMID: 37793345

Change to Text: Please kindly see the highlights (blue) in the revised manuscript. Line 156,163-166, 302-304, 396-399 and Figure S1.

Figure S1. The log-rank test grouping by high (top 25%) and low (bottom 25%) expression UBA1 in BC patients.

Kaplan-Meier survival curve showing the prognostic values of UBA1 expression in individuals with BC (A) overall survival; (B) disease-specific survival; and (C) progress-free survival

Comment 9: Providing the numbers for survival state is OK, but these should not be correlated with UBA1.

Authors’ Response: Thank you for your helpful suggestion. We have deleted the survival states in Table 1.

Change to Text: Please kindly see the Table 1 without the survival states in the revised manuscript. Table 1.

Round 3

Reviewer 2 Report (New Reviewer)

Comments and Suggestions for Authors

Thank you for this revision. I think you responded with sufficient changes in the manuscript, so that the manuscript can be published on my behalf.

This manuscript is a resubmission of an earlier submission. The following is a list of the peer review reports and author responses from that submission.

Round 1

Reviewer 1 Report

Comments and Suggestions for Authors

The authors have no intention to address the problems I mentioned last time. 

Here I repeat them again:

"Figures 1, 4, 5, and 6 are mostly repetitive with the Human Protein Atlas, which also uses TCGA and GTEx databases, showing all the interactive figures.

https://www.proteinatlas.org/ENSG00000130985-UBA1

https://www.proteinatlas.org/ENSG00000130985-UBA1/pathology/breast+cancer

https://www.proteinatlas.org/ENSG00000130985-UBA1/immune+cell

According to the Human Protein Atlas, UBA1 is not prognostic in breast cancer. The authors should explain why their conclusion is different when using the same datasets.

Figures 2 and 3 are redundant. GO BP analysis, KEGG analysis, and Hallmark analysis are all pathway analyses. Although the pathways from each analysis may have different names, they are not complementary to each other. It is redundant to include all of them unless you can explain in detail what extra information they provide.

The only novel findings are in Figures 7 and 8, which is not sufficient for a research article."

I will not say this is plagiarism since Figure 7 and 8 are original, but Figure 1-6 does not have any novelty. The content is overlapped with The Human Protein Atlas. The authors did not show any respect to other people's work, not even mentioning a single word about The Human Protein Atlas, neither citing any papers from the project.

Author Response

Reviewer 1

We are appreciated that you spend time evaluating our manuscript in your busy schedule. We made a point-by-point response to your comments. Please see the specific responses as follows:

Comment 1:

"Figures 1, 4, 5, and 6 are mostly repetitive with the Human Protein Atlas, which also uses TCGA and GTEx databases, showing all the interactive figures.

https://www.proteinatlas.org/ENSG00000130985-UBA1

https://www.proteinatlas.org/ENSG00000130985-UBA1/pathology/breast+cancer

https://www.proteinatlas.org/ENSG00000130985-UBA1/immune+cell

According to the Human Protein Atlas, UBA1 is not prognostic in breast cancer. The authors should explain why their conclusion is different when using the same datasets.

Authors’ Response: We acknowledge that Figures 1, 4, 5, and 6 in our manuscript exhibit a certain degree of overlap with the data presented in the Human Protein Atlas. It was indeed an oversight on our part not to adequately reference and discuss the existing research. To address this, we will explicitly cite the relevant studies from the Human Protein Atlas in our text and provide a detailed discussion on how our findings differ or offer novelty in comparison. Particularly concerning the role of UBA1 protein levels in the prognosis of breast cancer, we will conduct a comprehensive analysis of the datasets and elucidate the potential underlying reasons in our discussion.

Change to Text: Please kindly see the line 193-196 in the manuscript.

Comment 2:  Figures 2 and 3 are redundant. GO BP analysis, KEGG analysis, and Hallmark analysis are all pathway analyses. Although the pathways from each analysis may have different names, they are not complementary to each other. It is redundant to include all of them unless you can explain in detail what extra information they provide.

Authors’ Response: Thank you for your insights into the perceived redundancy of the GO BP, KEGG, and Hallmark analyses in our manuscript. We appreciate your attention to this matter and would like to clarify that each of these analyses serves a unique and complementary purpose in our research. The Gene Ontology Biological Process (GO BP) analysis provides a broad overview of the biological processes significantly enriched within our gene set, offering insights into the general functional categories that are active. This analysis is essential for identifying the overarching themes of gene regulation and function within the context of our experimental setup. Conversely, the Kyoto Encyclopedia of Genes and Genomes (KEGG) analysis focuses more intently on identifying specific pathways that are altered, which may encompass metabolic pathways, signaling cascades, and other defined processes. This analysis is invaluable for understanding the molecular mechanisms and interactions within the scope of our study. Lastly, the Hallmark analysis is particularly adept at recognizing coordinated expression patterns characteristic of certain biological states, such as cellular responses to stimuli or specific cell types. The uniqueness of this analysis lies in its ability to highlight signature expression profiles associated with well-defined biological processes.

While there may be some overlap in the information provided by these analyses, each offers a distinct perspective and depth of information that the others do not. Including all three analyses allows us to explore the multifaceted nature of the gene expression changes observed in our study comprehensively. To further address your concerns, we will enhance our discussion to clearly articulate the unique contributions of each analysis and how they collectively enhance our understanding of the data. We believe that the inclusion of all three analyses provides a more nuanced interpretation of our results.

We hope this explanation elucidates the significance of each analysis in our study and allays your concerns regarding redundancy. We are grateful for your feedback and remain committed to ensuring that our manuscript is rigorous and informative.

Change to Text: Please kindly see the line 218-223 in the manuscript.

Comment 3: Figure 1-6 does not have any novelty. The content is overlapped with The Human Protein Atlas. The authors did not show any respect to other people's work, not even mentioning a single word about The Human Protein Atlas, neither citing any papers from the project.

Authors’ Response: We acknowledge that the novelty of our manuscript is primarily concentrated in Figures 7 and 8. However, the bioinformatics section preceding these figures laid the necessary groundwork for our subsequent experimental validations. The potential of UBA1 in breast cancer, uncovered by our bioinformatics results, directed us to conduct further in vitro experiments for validation. The initial bioinformatics analysis was crucial in identifying UBA1 as a candidate worthy of more detailed investigation, which subsequently led to the innovative findings presented in Figures 7 and 8. We believe that this sequential approach from in silico to in vitro not only strengthens the credibility of our research findings but also provides a comprehensive perspective on the role of UBA1 in breast cancer.

In our ongoing and future research, we are committed to continuing this in-depth exploration from both in vivo and in vitro angles, and to unraveling the molecular mechanisms through which UBA1 exerts its effects in breast cancer. To enhance the research value of our paper, we have supplemented and verified the related results of TNF-α and IL-6 mRNA expression in MCF-7 cells treated with the E1 ubiquitin-activating enzyme inhibitor TAK-243 (MedChemExpress, Junction, NJ, USA, #HY-100487), as TNF-α and IL-6 are key initiators in the TNFα signaling and IL-17 signaling, as well as the IL-6-JAK-STAT3 pathways, which are significantly enriched based on our bioinformatics analysis. We found that the expression of both TNF-α and IL-6 was upregulated in MCF-7 cells when UBA1 was silenced (Figure 9). These results are consistent with our bioinformatics findings, further validating the role of UBA1 in the immune-related pathways associated with breast cancer.

We appreciate your feedback and are dedicated to ensuring that our work contributes meaningfully to the field.

Change to Text: Please kindly see line171-174, 290-295, 329-333, 221-226 and figure 9.

Reviewer 2 Report

Comments and Suggestions for Authors

The authors addressed the major issues of this reviewer.

Author Response

Thank you very much for your positive feedback on our revised manuscript. We are pleased to know that the revisions have adequately addressed the major issues raised by you. Your comments were instrumental in enhancing the quality of our manuscript and we have taken great care to incorporate them as faithfully as possible.